# Implicit Sense-labeled Connective Recognition as Text Generation

**Yui Oka** and **Tsutomu Hirao**

NTT Communication Science Laboratories, NTT Corporation

{yui.oka, tsutomu.hirao}@ntt.com

## Abstract

Implicit Discourse Relation Recognition (IDRR) involves identifying the sense label of an implicit connective between adjacent text spans. This has traditionally been approached as a classification task. However, some downstream tasks require more than just a sense label and the specific connective used. This paper presents Implicit Sense-labeled Connective Recognition (ISCR), which identifies the implicit connectives as well as their sense labels between adjacent text spans. ISCR can be treated as a classification task, but it's actually difficult due to the large number of potential categories, the use of sense labels, and the uneven distribution of instances among them. Accordingly, this paper instead handles ISCR as a text-generation task, using an encoder-decoder model to generate both connectives and their sense labels. Here, we explore a classification method and three types of text-generation methods. From our evaluation results on PDTB-3.0, we found that our classification method outperforms the conventional classification-based method.

## 1 Introduction

Discourse relations are often given adjacent text spans, such as clauses and sentences, without explicit connectives. Recognizing the implicit discourse relations (I.e., Implicit Discourse Relation Recognition (IDRR)) has been addressed as the main task of shallow discourse parsing. Figure 1 shows an example of IDDR. In PDTB-3.0 (Bonnie et al., 2019), a standard benchmark dataset for IDRR, 4 coarse-grained and 22 fine-grained discourse relations are defined between two text spans, referred to as Arg1 and Arg2. Recent studies have exploited pre-trained language models as vector representations for Arg1 and Arg2. In those cases, a model predicts a label of discourse relation by classifying the obtained vectors into classification layers. In other words, the previous stud-

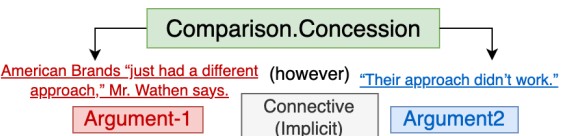

Figure 1: Example of IDRR. The discourse relation between Arg1 and Arg2 is `Comparison.Concession`. The corresponding connective provided by the annotator is *however*.

ies focused on identifying implicit discourse relations rather than implicit connectives. However, for downstream tasks such as machine translation, it may be necessary to handle both discourse relations and connectives (Meyer and Popescu-Belis, 2012; Li et al., 2014; Yang and Cardie, 2014). Here, connectives and discourse relations do not necessarily involve a one-to-one correspondence. When only a discourse relation is given, we must consider multiple candidate connectives corresponding to it. Furthermore, the candidate connective may already correspond to another discourse relation. Therefore, when exploiting IDRR in machine translation, incorrect translation may result from the failure to perform disambiguation of the connectives. This problem can also occur when only a single connective is provided. Figure 2 shows examples of such issues. This limitation has motivated us to identify both the connectives and their discourse relations, or sense-labeled connectives.

This paper addresses the task of predicting not only implicit discourse relation between text spans but also the connective, and we refer to this as Implicit Sense-labeled Connective Recognition (ISCR). ISCR is similar to IDRR in that it is also regarded as a classification task. However, ISCR is particularly challenging because it involves many classes in combining discourse relations and connectives. Moreover, the uneven distribution of instances belonging to classes also makes this task difficult. Therefore, inspired by Li et al. (2018), we regard our classification task as a text generation

This game console has been developed **first** for the Chinese market. ➡ 这款游戏机首先是为中国市场开发。
→**Expansion.Level-of-detail** →**Temporal.Asynchronous**

This game console has been developed **specifically** for the Chinese market. ➡ 这款游戏机是专门为中国市场开发的。
→**Expansion.Level-of-detail** →**Expansion.Level-of-detail**

Domestic car sales have plunged 19% **since** the Big Three ended many of their programs Sept. 30.
→**Contingency.Cause** ?
→**Temporal.Asynchronous**

Figure 2: Example of translations with ambiguous discourse relation and connective. A discourse relation, Expansion.Level-of-detail, corresponds to *specifically* and *first*, and then *first* also corresponds to Temporal.Asynchronous. A connective *since* corresponds to Contingency.Cause and to Temporal.Asynchronous.

task achieved by encoder-decoder models.

We explore three types of text generation to predict the implicit sense-labeled connectives: (1) Generating connectives with discourse relations only, (2) Generating connectives with discourse relations between Arg1 and Arg2, (3) Generating connectives between Arg1 and Arg2 and then predicting discourse relations based on a classifier using the connectives. The experimental results obtained from PDTB-3.0 demonstrate that generation-based approaches outperform a simple classification approach in terms of Accuracy, Macro-F1, and Weighted-F1.

## 2 Related Work

### 2.1 Implicit Discourse Relation Recognition (IDRR)

**Dataset**: PDTB-3.0 (Bonnie et al., 2019) is a recent standard benchmark dataset representing discourse relations between adjacent text spans using a hierarchical structure of three levels. The top level indicates the general category, such as Comparison, while the second level specifies the type, such as Concession. The third level provides further details about the relation, such as Arg2-as-denier. A period is used to distinguish each level, and thus for PDTB labels, a sequence of tokens such as Comparion.Concession.Arg2-as-denier is used. The most appropriate connectives to link Arg1 and Arg2 with a specific discourse relation are given as metadata.

**IDRR as Classification**: IDRR involves selecting the most appropriate discourse relation between Arg1 and Arg2 from the pre-defined relations. This process is carried out as a typical classification task. Usually, an end-to-end approach is adopted, where a large-scale language model is used to obtain vector representations of Arg1 and Arg2, which are then classified using a classification layer

(Shi and Demberg, 2019; Liu et al., 2020b; Xiang et al., 2022; Long and Webber, 2022). However, a large number of classes can lead to data sparseness, which is not ideal for classification-based approaches.

**IDRR via Connectives Prediction**: Generating connectives between Arg1 and Arg2 is easy for the current pre-trained encoder-decoder model. Therefore, predicting discourse relations with a classifier after generating connectives is feasible. Zhou et al. (2022) proposed methods that use prompt-tuning to generate connectives and then exploit classifiers to predict discourse relations from the generated connectives. Chan et al. (2023) viewed IDRR as a problem of predicting the hierarchical paths of connective and discourse relation labels, so they also proposed a prompt-tuning method. Jiang et al. (2023) proposed methods that learn the hierarchical discourse relation representations through multi-task learning and contrastive learning. Although they use connective prediction to identify the discourse relation between text spans, they do not give attention to the performance of connective identification. Furthermore, discourse relations are determined by using a classifier rather than using a text generator. Therefore, the performance of such methods would degrade as the number of classes to be predicted increases.

### 2.2 Classification as Text Generation

Simple classifier-based approaches face limitations when many classes with uneven distribution of instances are given. To tackle this difficulty, Li et al. (2018), Kwon et al. (2023), and Wu et al. (2021) proposed a novel approach based on text generation by pre-trained encoder-decoder models. For a hierarchical text classification task, Kwon et al. (2023) used a generation-based classifier to capture the label hierarchy and unseen labels explicitly. Wu et al.

(2021) viewed IDRR as a problem of predicting only the sequence of discourse labels. Although this method predicts sequences containing a connective, the connective plays only an auxiliary role in predicting discourse relations, and this method does not focus on predicting the connective.

Using a pre-trained encoder-decoder enables us to generate class labels that cannot be found, or are rare, in the training data. This is possible because the datasets used for pre-training may contain such class labels, which implies a significant benefit of using text generation as a classification technique.

## 3 Text Generation for ISCR

In ISCR, the number of sense-labeled connectives to be predicted is significantly larger than that of IDRR, and the distribution of instances belonging to each class is unbalanced. The label of the ISCR task is composed of a discourse relation label and connective words. Considering that it is not difficult to generate connective words with the pre-trained encoder-decoder model, we employ a text generation approach following Li et al. (2018) and Kwon et al. (2023).

One possible way of improving the ISCR task is to incorporate contextual information obtained from Arg1 and Arg2 in the encoder-decoder model. Therefore, we examined the following three generation methods using encoder-decoder models.

**Connective-Label Generation (CLG)**: Connectives and discourse relation labels, or 'Connective (Discourse Relation Label),' are generated using an encoder-decoder model such as Li et al. (2018) (Figure 3, left). The reason for using the expression 'Connective (Discourse Relation Label)' is that the training data for the pre-trained encoder-decoder model contain many expressions 'some expression + (its description).' When the decoder outputs a parenthesis after a particular word, we expect it to predict a description of the word.

**CLG plus Preceding and Following text span (CLG+PF)**: Discourse relation labels and connectives with Arg1 and Arg2, 'Arg1 Connective (Discourse Relation Label) Arg2,' are generated (Figure ??). We believe that the prediction of discourse relation labels and connectives with Arg1 and Arg2 is easier for encoder-decoder models.

**Classification after Connective Generation (CCG)**: Connectives are generated and then discourse relation labels are predicted using a simple classifier with a tuple of Arg1, connective, Arg2 (Figure 3, right). This is a simplified variant of prompt-based methods.

## 4 Experiment Setting

**Dataset**: We used the Penn Discourse TreeBank version 3.0 (PDTB-3.0) (Bonnie et al., 2019) [1] to evaluate the performance of ISCR. This dataset contains annotated discourse relations in articles from the Wall Street Journal. While they have relation labels at three levels, this study used labels only up to the second level, following the previous IDRR studies (Ji and Eisenstein, 2015; Xiang et al., 2022; Long and Webber, 2022). In addition, connective words provided by annotators were used.

Here, we also used the standard train/val/test split of the dataset (Ji and Eisenstein, 2015). Training data consisted of Sections 2-20, validation data used Sections 0-1, and testing used Sections 21-22. The PDTB-3.0 dataset contains 99 ISCR classes at the top level and 133 at the second level in test set, while there are only four classes at the top level and 17 at the second level for IDRR.[2] Since ISCR predicts sense-labeled connectives, the number of classes increases, which results in unbalanced data. [3]

**Parameter Settings**: We used BART-base (Lewis et al., 2020) as the encoder-decoder model and RoBERTa-base (Zhuang et al., 2021) as the simple classifier. For optimization, we used RAdam (Liu et al., 2020a). The learning rate for the classifier and text-generation model was set to 1e-6, with a batch size of 4. We conducted training for 30 epochs and implemented early stopping after 3.

**Evaluation Metrics**: In IDRR, Accuracy and Macro-F1 are commonly used as metrics for evaluation. However, in ISCR, the number of sense-labeled connectives is significantly larger than that in IDRR, and the unbalanced data distribution is more pronounced. Therefore, we used Weighted-F1, which is used as an evaluation metric for unbalanced data distribution.

## 5 Results and Discussion

**Results of ISCR**: Table 1 shows the results for ISCR. The Vanilla Classifier is a simple classification approach using RoBERTa (Zhuang et al.,

---

[1]https://catalog.ldc.upenn.edu/LDC2019T05

[2]The original label of PDTB-3 is excluded.

[3]For more information on the label distributions of IDRR and ISCR, please see the Appendix.

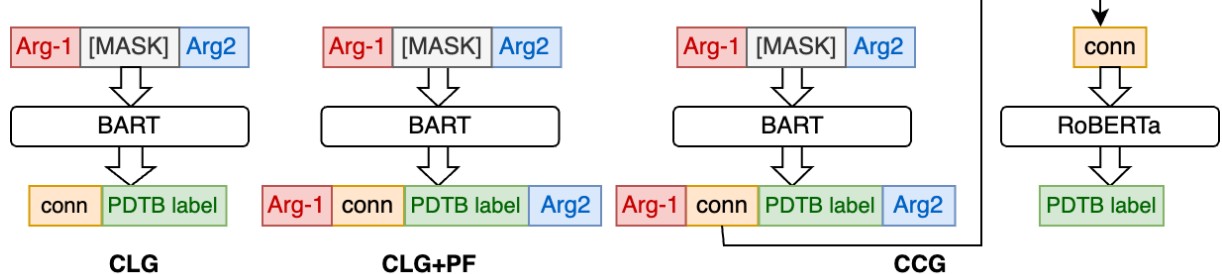

Figure 3: Overview of proposed method

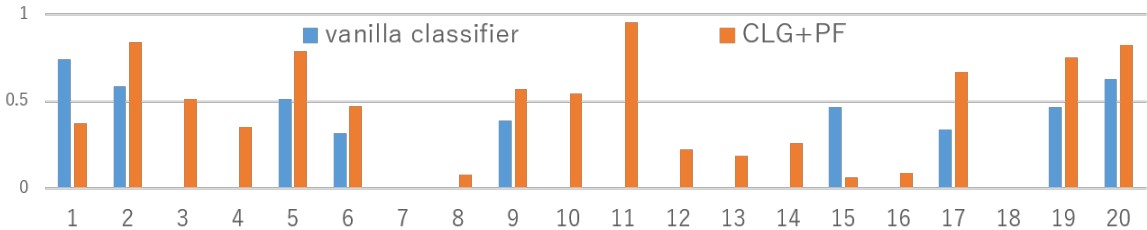

Figure 4: F1 score (%) for each label at top level. 1. also (expansion), 2. and (expansion), 3. as a result (contingency), 4. as (contingency), 5. because (contingency), 6. but (comparison), 7. by comparison (comparison), 8. for instance (expansion), 9. however (comparison), 10. in fact (expansion), 11. in order (contingency), 12. in other words (expansion), 13. in particular (expansion), 14. indeed (expansion), 15. instead (expansion), 16. meanwhile (temporal), 17. so (contingency), 18. since (contingency), 19. specifically (expansion), 20. then (temporal).

|  | Model | Acc | M-F1 | W-F1 |
|---|---|---|---|---|
| Top | Vanilla Classifier | 33.24 | 7.78 | 32.83 |
|  | CLG | 38.55 | 16.55 | 35.14 |
|  | CLG+PF | 39.75 | 20.06 | 48.67 |
|  | CCG | 39.80 | 20.86 | 45.65 |
| Second | Vanilla Classifier | 31.95 | 5.04 | 31.15 |
|  | CLG | 36.63 | 12.17 | 33.01 |
|  | CLG+PF | 38.80 | 16.21 | 47.32 |
|  | CCG | 38.80 | 15.93 | 47.31 |

Table 1: Results for ISCR

|  | Model | Acc | M-F1 | W-F1 |
|---|---|---|---|---|
| Top | (Xiang et al., 2022) | 64.04 | 56.63 | - |
|  | (Long and Webber, 2022) | 75.31 | 70.05 | - |
|  | Vanilla Classifier | 71.52 | 66.81 | 71.43 |
|  | CLG+PF | 71.66 | 65.74 | 71.61 |
| Second | (Long and Webber, 2022) | 64.68 | 57.62 | - |
|  | Vanilla Classifier | 61.66 | 50.39 | 60.41 |
|  | CLG+PF | 60.68 | 50.33 | 59.47 |

Table 2: Results for IDRR

| Task | Acc | M-F1 | W-F1 |
|---|---|---|---|
| Vanilla Classifier | 37.65 | 12.26 | 33.47 |
| CLG+PF | 43.99 | 28.54 | 52.42 |

Table 3: Results for connective prediction

dicate the effectiveness of generating the connective+label along with the preceding and following text spans. The difference is more enhanced for Macro-F1 and Weighted-F1, while CLG+PF and CCG show almost the same score for accuracy.

To discuss the effectiveness of CLG+PF in more detail, we show F1 scores of the Vanilla Classifier and CLG+PF for each label in Figure 4. Note that only labels that occur more than 20 times in the test set are listed. While the Vanilla Classifier does not predict many labels, CLG+PF can predict labels in most cases.

**Results of IDRR**: Table 2 shows evaluation results for IDRR. We found that the Vanilla Classifier performs better than that of Xiang et al. (2022), while it is inferior to the method of Long and Webber (2022). This is because Xiang et al. (2022) used BERT (Devlin et al., 2019), which seems inferior to RoBERTa. CLG+PF is also inferior to Long and Webber (2022), but it achieves comparable performance to the Vanilla Classifier. Furthermore,

2021) that combines individual connectives and discourse relations prediction. First, when comparing our methods with the Vanilla Classifier, our methods obtained better results for all evaluation metrics. These results suggest the effectiveness of classification as a text-generation approach. Second, when comparing our methods on Weighted-F1, CLG+PF is better than CLG. The results in-

CLG+PF outperformed the method of Xiang et al. (2022).

**Results of Connective Prediction** The results of connective prediction by CLG+PF and the Vanilla Classifier are given in Table 3. CLG+PF outperformed the Vanilla Classifier for all evaluation metrics with large differences. These results are unsurprising, since connective prediction is more complicated than IDRR due to the larger number of classes.

## 6 Conclusion

This paper investigated ISCR as a way to predict implicit sense-labeled connectives between adjacent text spans. Here, we leveraged a text-generation approach, using encoder-decoder models, rather than a simple classification approach. This strategy was motivated by the large number of sense-labeled connectives and their unbalanced distribution. Experimental results obtained from PDTB-3.0 show that our method is superior to the simple classification method. When generating connectives and corresponding relation labels together with Arg1 and Arg2, the proposed method achieved the best performance on Weighted-F1. This method obtained comparable performance in the IDRR setting and better performance for connective prediction than a simple classification approach.

## Limitations

Our approach is suitable only for English ISCR because it necessitates pre-trained encoder-decoder models and sufficiently large training datasets for fine-tuning the models. To apply our approach to other languages, we would need datasets in those languages that describe the correspondences between connectives and their discourse relations. We assume that connectives are placed between Arg1 and Arg2, and actually there are only a few cases in English where this assumption is not supported. It remains unclear whether this assumption is necessarily valid for other languages.

Newly introduced discourse relations such as +speech and +belief in PDTB-3.0 are difficult to predict using only features obtained from a text. In order to predict such discourse relations, we need to obtain the implied intent and confidence level of speakers. How to acquire such information remains a significant challenge not only for shallow discourse parsing but also for NLP.

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

# A  Appendix

| level | label | sample |
|---|---|---|
| top | comparison | 154 |
| | contingency | 529 |
| | expansion | 643 |
| | temporal | 148 |
| second | comparison.concession | 98 |
| | comparison.contrast | 54 |
| | comparison.similarity | 2 |
| | contingency.cause | 406 |
| | contingency.cause+belief | 15 |
| | contingency.cause+speechact | 4 |
| | contingency.condition | 15 |
| | contingency.purpose | 89 |
| | expansion.conjunction | 236 |
| | expansion.disjunction | 2 |
| | expansion.equivalence | 30 |
| | expansion.instantiation | 124 |
| | expansion.level-of-detail | 208 |
| | expansion.manner | 17 |
| | expansion.substitution | 26 |
| | temporal.asynchronous | 105 |
| | temporal.synchronous | 43 |

Table 4: Number of samples for each label in PDTB-3.0

| | sample |
|---|---|
| because (contingency), and (expansion) | 113 |
| for instance (expansion) | 109 |
| specifically (expansion) | 100 |
| so (contingency) | 87 |
| in order (contingency) | 86 |
| then (temporal) | 80 |
| as a result (contingency) | 71 |
| in fact (expansion) | 45 |
| but (comparison) | 41 |
| however (comparison) | 40 |
| as (contingency), in other words (expansion) | 33 |
| in particular (expansion) | 32 |
| also (expansion) | 23 |
| indeed (expansion), instead (expansion) | 22 |
| by comparison (comparison), meanwhile (temporal) | 21 |
| since (contingency), thus (contingency) | 20 |
| while (temporal) | 18 |
| although (comparison), consequently (contingency) | 14 |
| while (expansion), therefore (contingency) | 13 |
| by (expansion), meanwhile (expansion) | 12 |
| moreover (expansion) | 10 |
| accordingly (contingency), for example (expansion), in short (expansion) | 9 |
| as a result of being (contingency), in addition (expansion), whereas (comparison) | 8 |
| in contrast (comparison), further (expansion), subsequently (temporal) | 7 |
| that is (expansion), but (expansion), by contrast (comparison), furthermore (expansion), when (contingency), previously (temporal) | 6 |
| rather (expansion), in (expansion), on the whole (expansion) | 5 |
| as a result of (contingency), because of (contingency), besides (expansion), even though (comparison), nevertheless (comparison) | 4 |
| given (contingency), similarly (expansion), thereby (expansion), when (temporal) | 3 |
| eventually (temporal), at that time (temporal), if it is (contingency), if they were (contingency), in short (contingency), more specifically (expansion), before (temporal), first (expansion), nonetheless (comparison), however (expansion), or (expansion), while (comparison), with (expansion), with the purpose of (contingency), yet (comparison), overall (expansion), similarly (comparison), so as (contingency) | 2 |
| as (temporal), as it turns out (expansion), as such (contingency), at the same time (temporal), earlier (temporal), hence (contingency), if (contingency), if one is (contingency), if they are (contingency), if you are (contingency), if you were (contingency), in return (expansion), in sum (expansion), in that (expansion), in the end (temporal), it is because (contingency), later (temporal), likewise (expansion), next (temporal), second (expansion), separately (expansion), so (expansion), third (expansion), though (comparison), ultimately (temporal) | 1 |

Table 5: Each label at the top level of ISCR and the number of samples for each label