# OpenReview forum: "Implicit Sense-labeled Connective Recognition as Text Generation"
_EMNLP/2023/Conference — EMNLP 2023 Findings_

### Official Review · Reviewer_ZGAK · 2023-08-01

**Soundness:** 3

**Excitement:**

2: Mediocre: This paper makes marginal contributions (vs non-contemporaneous work), so I would rather not see it in the conference.

**Paper Topic And Main Contributions:**

This paper focuses on Implicit Sense-labeled Connective Recognition (ISCR), which identifies both the sense labels and connectives between adjacent text spans. Three text-generation based models are adapted to this task and achieve promising results on PDTB3.0.

**Questions For The Authors:**

1) In my opinion, ISCR is a special case of multi-level implicit discourse relation recognition [1][2]. It would be better to discuss the key differences between them.
[1] Global and Local Hierarchy-aware Contrastive Framework for Implicit Discourse Relation Recognition, Findings of ACL 2023 or Earlier Version on arXiv.
[2] A Label Dependence-Aware Sequence Generation Model for Multi-Level Implicit Discourse Relation Recognition, AAAI 2022.

2) In Figure 3, CCG model is somewhat confusion because only conn is feed into the RoBERTa-based classification model.

**Reasons To Accept:**

1) The used generation-based models are effective on ISCR, especially when the label distribution is uneven.

**Reasons To Reject:**

1) This paper is the direct application of existing methods on the similar task. The used generation-based models [Kwon et al. (2023)] are proposed for hierarchical text classification, which is the similar task as ISCR.

**Reproducibility:**

4: Could mostly reproduce the results, but there may be some variation because of sample variance or minor variations in their interpretation of the protocol or method.

**Reviewer Confidence:**

4: Quite sure. I tried to check the important points carefully. It's unlikely, though conceivable, that I missed something that should affect my ratings.

---

> ### Author Rebuttal · Authors · 2023-08-29
>
> Thanks for your constructive comments and questions, and they are exceedingly helpful to improve our paper.
>
> First of all, our proposed ISCR differs from [Kwon et al. (2023)]. In [Kwon et al. (2023)], a single text was input and a generative model predicted the hierarchical labels that applied to that sentence. On the other hand, our task is to predict both the relationship between two texts and the connectives that could be inserted between them. Note that the problem setup and the generative model work differently. In the proposed method, the sequence of Arg1, connectives, PDTB labels, and Arg2 is trained to be a natural. Therefore, it differs from methods that simply generate hierarchical labels.
>
> 1. Our focus is not on estimating discourse relation labels, but rather on estimating appropriate unambiguous connectives. Multilevel IDRR, on the other hand, focuses on the estimation of discourse relation labels. This is a major difference.
> In fact, the connective prediction in [1] is 19.2 (Macro-F1*), while our method is 28.5. In other words, our method has a significant advantage over [1] in connective prediction.
> In [2], only the PDTB-2 was evaluated, so it cannot be discussed. However, the results seem to indicate that the connective classification scores are low. Similar to [1], [2] focuses on the prediction of discourse relation labels. As for performance, [2] only reports experimental results for PDTB-2, so no comparison can be made. However, because of its focus on discourse relation label prediction, we believe that its performance on connective prediction is lower.
>
>
> 2. We proposed to investigate whether it is better to generate PDTB labels with connectives or to predict PDTB labels from connectives.However, we believe that learning to generate PDTB labels along with connectives is better for disambiguating connectives than learning to generate only connectives. Therefore, we trained the generative model to output the connective along with the PDTB label.
>
>
> *PDTB dataset is unbalanced and will be discussed in Macro-F1.
>
> [1] Global and Local Hierarchy-aware Contrastive Framework for Implicit Discourse Relation Recognition, Findings of ACL 2023 or Earlier Version on arXiv.
>
> [2] A Label Dependence-Aware Sequence Generation Model for Multi-Level Implicit Discourse Relation Recognition, AAAI 2022.

---

### Official Review · Reviewer_jhFv · 2023-08-02

**Soundness:** 3

**Excitement:**

4: Strong: This paper deepens the understanding of some phenomenon or lowers the barriers to an existing research direction.

**Missing References:**

Some relevant literature:
- Annotation-Inspired Implicit Discourse Relation Classification with Auxiliary Discourse Connective Generation: https://arxiv.org/abs/2306.06480
- Analysis between connective and relation: https://aclanthology.org/2021.codi-main.7.pdf
- Agreements between gold and predicted connectives: https://aclanthology.org/2021.unimplicit-1.1.pdf
- Generating Discourse Connectives with Pre-trained Language Models:
Conditioning on Discourse Relations Helps Reconstruct the PDTB https://aclanthology.org/2022.sigdial-1.48.pdf


**Paper Topic And Main Contributions:**

This paper presents a generation-based encoder-decoder approach to predict both implicit discourse connectives and relations (i.e. Implicit Sense-labeled Connective Recognition, ISCR).
They propose 3 generation-based approaches and show that all of them perform better than a vanilla classifier on ISCR and connective prediction.
However, their combined approach fails to outperform SOTA on the traditional IDRR task.

**Reasons To Accept:**

The observation that discourse relations and connectives form a many-to-many relation and the motivation to predict both in a discourse model is forward-looking.
In addition to reporting performances on the popular Implicit Discourse Relation Recognition (IDRR) task, this paper presents results in implicit connective prediction as well as ISCR.
Figure 4 shows great potential for better macro-level predictions on implicit connective and relation when compared to a vanilla classifier.



**Reasons To Reject:**

Even though constructing ISCR as a generation task and including both connective and sense in prediction advocates a more comprehensive discourse model, this paper's performance on the traditional IDRR task does not outperform previous SOTA.


Some concerns:
- It would be more helpful to present both level1 and level2 performances in Table 2
- Does the training set provide all possible combinations of implicit relations X connectives? if not, results/analyses on unattested relation X connective combinations would be useful
- Figure 4 could include test frequencies of the top 20 connective X relation combinations


**Reproducibility:**

4: Could mostly reproduce the results, but there may be some variation because of sample variance or minor variations in their interpretation of the protocol or method.

**Reviewer Confidence:**

4: Quite sure. I tried to check the important points carefully. It's unlikely, though conceivable, that I missed something that should affect my ratings.

---

> ### Author Rebuttal · Authors · 2023-08-29
>
> Thanks for your constructive comments and questions, and they are exceedingly helpful to improve our paper.
> We will respond to your concerns.
> ***
>
> - It would be more helpful to present both level1 and level2 performances in Table 2
>
> Ans: Thank you for pointing that out. level-2 results are not included due to space limitations. Level-2 results will be listed in the appendix. Level-2 results showed the same trend as level-1.
> |  model |  Acc  | M-F1  | W-F1  |
> | ---- | ---- | ---- | ---- |
> |  vanilla classifier  |  61.66  |  50.39  |  60.41  |
> |  CLG+PF  |  60.68  |  50.33 8|  59.47  |
>
> ***
> - Does the training set provide all possible combinations of implicit relations X connectives? if not, results/analyses on unattested relation X connective combinations would be useful
>
> Ans: No, it does not.The PDTB dataset comes with implicit PDTB labels and possible connectives terms as metadata. We train on all of this metadata.It is possible to try all possible combinations and to expand the data, but observing the combination of PDTB labels and connectives, we decided it was better not to do so. For example, for 'expansion.conjunction' the following connectives are candidates.
> ```
> ['and', 'indeed', 'but', 'furthermore',  'first', 'second', 'third', 'meanwhile', 'however', 'moreover', 'in addition', 'besides',  'also',  'overall', 'further',  'in fact', 'as it turns out', 'separately', 'while', 'likewise', 'in return', 'similarly', 'in short', 'in other words',  'that is', 'on the whole']
> ```
> If all these possibilities are learned, they are considered noise. For example, 'first' and 'second' have different meanings. Furthermore, by PDTB label, 'expansion.disjunction' and 'expansion.equivalence' contain only 2-5 conjunctions, and the number of connectives included varies widely by PDTB label. Therefore, we do not learn all possible combinations.
>
>
> ***
> - Figure 4 could include test frequencies of the top 20 connective X relation combinations
>
> Ans: Thank you for your suggestion, we will add to the text about the frequency. They are as follows respectively.
> ```
> also (expansion)  23, and (expansion) 113, as a result (contingency)  71, as (contingency)  33, because (contingency) 133, but (comparison) 41, by comparison (comparison)  21, for instance (expansion) 109, however (comparison)   40, in fact (expansion) 45, in order (contingency)  86, in other words (expansion) 33, in particular (expansion)  32, indeed (expansion)  22, instead (expansion)  22, meanwhile (temporal)  21, so (contingency) 87, since (contingency)  20, specifically (expansion) 100, then (temporal) 80
> ```

---

### Official Review · Reviewer_mtYh · 2023-08-03

**Typos Grammar Style And Presentation Improvements:** The writing except the Abstract needs…
**Soundness:** 3

**Excitement:**

2: Mediocre: This paper makes marginal contributions (vs non-contemporaneous work), so I would rather not see it in the conference.

**Missing References:**

No

**Paper Topic And Main Contributions:**

This paper proposes a new task called Implicit Sense-labeled Connective Recognition (ISCR). And the authors devise a generation model to solve it. My main concerns line in the motivations of this task. Also, I am not convinced the point that generation models are more suitable than classification models for uneven label distribution.

**Questions For The Authors:**

Questions:
- 1. I don't understand your motivation clearly, especially in lines 52-54. Why is the ambiguity of connective related to your new ISCR task? In my opinion, it should be more related to the word disambiguation task. In this case, each connective can be labeled as mean1, mean2, and so on.
- 2. Is there any paper that proves the generation model performs better than the classification model when the number of labels increases? You mentioned that the method is inspired by paper[1][2]. But you should know those papers are working in hierarchical classification tasks, in which the correlation information between labels is helpful. They used a generation model to capture such correlation information because the fine-grained label generated at the latter steps can use the coarse label generated at the early steps. But I didn't see your clear motivation to use generation models.
- 3. The description on lines 129-133 is misleading. These two papers address difficulties in hierarchical classification tasks where a seq2seq model can capture correlations between labels. But in lines 129-133, the author describes that they are designed for uneven distribution of instances.
- 4. The CCG in Figure 3 first generates Conn and PDTB label then predicts the PDTB label. I am curious why you need to predict the PDTB label again after you already got it in step one.
- 5. Is the size of 99 and 133 on lines 194 and 195 correct? Because I see there are more than 200 connectives in PDTB 3.0.

References:
- [1] Maggie Yundi Li, Stanley Kok, Liling Tan. Don’t Classify, Translate: Multi-Level E-Commerce Product Categorization Via Machine Translation. In CoRR, abs/1812.05774.
- [2] Jingun Kwon, Hidetaka Kamigaito, Young-In Song, Manabu Okumura. Hierarchical Label Generation for Text Classification. In Findings of EACL 2023.


**Reasons To Accept:**

- The abstract of this paper is very clear.
- The example in Figure 2 is interesting.
- The proposed method shows better performance than the baseline.

**Reasons To Reject:**

- The motivation is not very clear (see questions).
- The setting in experiments seems not correct. For example the total number of labels in PDTB 3.0 (see questions).

**Reproducibility:**

3: Could reproduce the results with some difficulty. The settings of parameters are underspecified or subjectively determined; the training/evaluation data are not widely available.

**Reviewer Confidence:**

4: Quite sure. I tried to check the important points carefully. It's unlikely, though conceivable, that I missed something that should affect my ratings.

---

> ### Author Rebuttal · Authors · 2023-08-29
>
> Thanks for your constructive comments and questions, and they are exceedingly helpful to improve our paper.
>
>
> 1. Our ISCR proposal is not a word disambiguation task.In a word disambiguation task, the ambiguous word is included in the input sentence, whereas in ISCR, the connective is not given as it is implicit in the input.
>
>
> 2. Learning to generate connectives that naturally fall between Arg1 and Arg2 is the motivation for our method. Our goal is to use something other than a hierarchical structure. The reasons for our expectations for the generative models are described in L153-171. Please reread it, as the expected effect is different for each model.
>
>
> 3. Thank you for pointing this out. Indeed, these papers approach the hierarchical label classification task using the seq2seq model. On the other hand, the distribution of labels could be more balanced at the lower levels of these hierarchical label-based classification tasks. Even in the PDTB dataset, the data is unbalanced at level 2. Therefore, these papers can be viewed as an approach to unbalanced data.
> *For reference, we describe the distribution of level 2 in the PDTB-3 training set at the end of the comments.
>
>
> 4. We proposed this to investigate whether it is better to generate PDTB labels together with connectives or whether it is better to predict PDTB labels from connectives. However, we believe that learning to generate PDTB labels together with connectives will lead to disambiguation of connectives, rather than learning to generate only connectives. Therefore, we trained the generative model to output connectives and PDTB labels.
>
>
> 5. Thank you for pointing this out, and we are very sorry for the misunderstanding. You are correct; the connectives listed in PDTB are without the original PDTB-3.0 labels, which were excluded in this study. The second-level results will be presented in the Appendix, and the second-label scores will be compared with the Long et al. scores, so the original PDTB3.0 labels were omitted. However, this description was not included in the main text, so it is added in the footnote.
>
> *The distribution of level 2 in the PDTB-3 training set. We could list the distribution of connectives + PDTB labels, but due to the large number of labels, we will show level-2.
>
> ```
> {'expansion.conjunction': 236, 'contingency.cause.result': 210, 'contingency.cause.reason': 196, 'expansion.level-of-detail.arg2-as-detail': 191, 'expansion.instantiation.arg2-as-instance': 124, 'temporal.asynchronous.precedence': 94, 'contingency.purpose.arg2-as-goal': 89, 'comparison.concession.arg2-as-denier': 87, 'comparison.contrast': 54, 'temporal.synchronous': 43, 'expansion.equivalence': 30, 'expansion.substitution.arg2-as-subst': 26, 'expansion.level-of-detail.arg1-as-detail': 17, 'contingency.condition.arg2-as-cond': 15, 'expansion.manner.arg2-as-manner': 13, 'temporal.asynchronous.succession': 11, 'comparison.concession.arg1-as-denier': 11, 'contingency.cause+belief.result+belief': 8, 'contingency.cause+belief.reason+belief': 7, 'expansion.manner.arg1-as-manner': 4, 'contingency.cause+speechact.reason+speechact': 3, 'expansion.disjunction': 2, 'comparison.similarity': 2, 'contingency.cause+speechact.result+speechact': 1}
> ```

---

### Meta-Review · Area_Chair_sjy1 · 2023-09-19

**Recommendation:** 3

**Metareview:**

This paper tackles the task of Implicit Sense-labeled Connective Recognition (ISCR) as a text-generation task by proposing an encoder-decoder model to generate both the connective and their sense labels while previous work has been addressing the problem as a classification task. The authors examined the relation between connectives and their corresponding sense labels and presented promising results of the proposed generation-based approach to the task. During the rebuttal period, the authors provided further details to address reviewers’ comments and questions regarding the implications of label distribution and hierarchical label classification task.

The paper would benefit from including results from level 2 labels as well as relevant data insights that can contribute to a better understanding of the results of the task in relation to the proposed approach, as mentioned by the reviewers (jhFv and mtYh).

---

### Decision · Program_Chairs · 2023-10-07

**Decision:**

Accept-Findings

**Comment:**

This paper tackles the task of Implicit Sense-labeled Connective Recognition (ISCR) as a text-generation task by proposing an encoder-decoder model to generate both the connective and their sense labels while previous work has been addressing the problem as a classification task. The authors examined the relation between connectives and their corresponding sense labels and presented promising results of the proposed generation-based approach to the task. During the rebuttal period, the authors provided further details to address reviewers’ comments and questions regarding the implications of label distribution and hierarchical label classification task.

The paper would benefit from including results from level 2 labels as well as relevant data insights that can contribute to a better understanding of the results of the task in relation to the proposed approach, as mentioned by the reviewers (jhFv and mtYh).